# Seeing the Forrest through the trees: Oculomotor metrics are linked to heart rate

**Alex J. Hoogerbrugge**[1]*, **Christoph Strauch**[1], **Zoril A. Oláh**[1], **Edwin S. Dalmaijer**[2], **Tanja C. W. Nijboer**[1,3,4], **Stefan Van der Stigchel**[1]

**1** Experimental Psychology, Helmholtz Institute, Utrecht University, Utrecht, Netherlands, **2** School of Psychological Science, University of Bristol, Bristol, United Kingdom, **3** Center of Excellence for Rehabilitation Medicine, UMC Utrecht Brain Center, University Medical Center Utrecht, De Hoogstraat Rehabilitation, Utrecht, Netherlands, **4** Department of Rehabilitation, Physical Therapy Science & Sports, UMC Utrecht Brain Center, University Medical Center Utrecht, Utrecht, Netherlands

* a.j.hoogerbrugge@uu.nl

**Data Availability Statement:** Links to the raw data and their descriptions can be retrieved via https://studyforrest.org (Hanke et al., 2016 [https://doi.org/10.1038/sdata.2016.92] for the original

## Abstract

Fluctuations in a person's arousal accompany mental states such as drowsiness, mental effort, or motivation, and have a profound effect on task performance. Here, we investigated the link between two central instances affected by arousal levels, heart rate and eye movements. In contrast to heart rate, eye movements can be inferred remotely and unobtrusively, and there is evidence that oculomotor metrics (i.e., fixations and saccades) are indicators for aspects of arousal going hand in hand with changes in mental effort, motivation, or task type. Gaze data and heart rate of 14 participants during film viewing were used in Random Forest models, the results of which show that blink rate and duration, and the movement aspect of oculomotor metrics (i.e., velocities and amplitudes) link to heart rate–more so than the amount or duration of fixations and saccades. We discuss that eye movements are not only linked to heart rate, but they may both be similarly influenced by the common underlying arousal system. These findings provide new pathways for the remote measurement of arousal, and its link to psychophysiological features.

## Introduction

Remotely and unobtrusively detecting fluctuations in arousal is of wide interest to researchers in fields such as human-computer interaction, psychology, and ergonomics. This interest is due to the fact that changes in arousal are not only related to physical exertion, but also to psychological concepts for which arousal is often assessed as an objective approximation, such as the degree of excitedness, drowsiness, or mental effort during a given task. Given that arousal levels are related to task performance following an inverted U-shape function [1, 2], they have a profound effect on task performance–for instance on various critical tasks, arousal may affect the safety of operators and other people who rely on those operators [3]. Although fluctuations in arousal can be detected from various objective sources such as electroencephalography (EEG), functional Magnetic Resonance Imaging (fMRI), heart rate, or skin conductance [4], these methods require direct physical interaction with measurement devices or can be quite obtrusive. Only few parameters can be assessed remotely, such as oculomotor metrics obtained via video-based eye-tracking.

publication and OpenNeuro [https://openneuro.org/datasets/ds000113/versions/1.3.0] for the direct link to the data).

**Funding:** This work was supported by ERC [ERC-CoG-863732], https://erc.europa.eu/, awarded to SVdS. The funders had no role in study design, data collection and analysis, decision to publish, or preparation of the manuscript.

**Competing interests:** The authors have declared that no competing interests exist.

In the current study we investigate how well heart rate–one of the best investigated central indicators of arousal–can be predicted from remotely accessible oculomotor metrics as alternative peripheral indicators of arousal. A link between these indicators is plausible given the extensive support for correlations between oculomotor metrics and various psychological concepts, such as mental effort. For instance, it has been shown that the degree of pupil dilation can provide an accurate indication of participants' mental effort in both controlled and naturalistic viewing tasks [5, 6]. Furthermore, it has been shown that the peak velocity of saccades decreases as mental effort increases [7, 8] and increases as motivation increases [9]. Similarly, mental effort has been shown to covary with heart rate and with several derivatives of heart rate measures [10]. Additionally, it has been shown that changes in arousal are paired with an altered rate of eyeblinks [11, 12], and that spontaneous eyeblinks occur in tandem with an increase in heart rate variability [13].

While oculomotor measures are fairly robust, they can be influenced by the environmental circumstances under which they were obtained. For instance, pupil dilation is impacted by the luminance of the scene that is being watched, and microsaccades and peak velocities of saccades can only be reliably measured by expensive high-speed, low-noise trackers. Additionally, among eye tracking scientists there is no unified concept of how fixations and saccades should be defined–and thus the application of differing fixation- and saccade detection techniques may result in differing outcomes, even if they are applied to the same dataset [14]. As such, incorporating several metrics which can be independently extracted (e.g., pupil size, oculomotor movement, blinks) would improve robustness of the model, as it reduces dependence on one single extraction technique. This also applies to cases in which pupil dilation measurements are unreliable or missing, or the eye tracker's sampling rate is too low to extract peak saccade velocities.

The benefits of relating oculomotor metrics to heart rate are two-fold. Firstly, scrutinizing the links between oculomotor metrics and heart rate can foster our theoretical understanding of common underlying mechanisms and thereby our definition of arousal. Secondly, ever since the seminal works of Buswell [15] and Yarbus [16], we have been aware that eye movements are foremost driven by task type (top-down) and visual saliency (bottom-up). Later on, it has been shown more reliably that task type–such as search or free-viewing–influences gaze behaviour [17–19], and that oculomotor metrics besides pupil dilation [20] and peak saccade velocity (e.g., saccade amplitude, fixation duration) can provide sufficient information for machine learning algorithms to predict task type at above chance level [21, 22]. In this manuscript, we describe that, besides top-down and bottom-up mechanisms, arousal–estimated by the link to heart rate–also contributes to eye movements.

To this end, we use data from the *studyforrest* dataset [23]. This dataset contains eye tracking and pulse oximetry measurements from participants while they watched the 1994 motion picture *Forrest Gump*. We investigate whether oculomotor metrics can provide sufficient information for regression models and machine learning models to accurately predict high or low heart rates of participants in this naturalistic viewing task. Furthermore, we investigate how strongly each oculomotor feature contributes to the correct prediction of heart rate, thereby providing insight into how specific aspects of oculomotor movement is driven by a common measurement of arousal, such as heart rate.

## Methods

All analyses were performed with Python 3.8.10, using SciPy version 1.6.2 and scikit-learn version 0.24.2 [24, 25]. All code and outcomes can be retrieved from https://osf.io/skcd8/.

### Raw data

Eye tracking data and pulse oximetry data were obtained from the *studyforrest* dataset, which contains data of fourteen participants that were measured while being presented with the 2-hour film *Forrest Gump* [23, 26]. The raw eye tracking data was measured with an Eyelink 1000 at a frequency of 1 kHz and pulse oximetry measurement was applied to record heart rate data at an effective frequency of 100 Hz. A full description of the recordings and anomalies can be found in [23] and at https://studyforrest.org.

### Oculomotor feature detection

Fixations and saccades were extracted based on the algorithm proposed in [27], which operationalizes fixations and saccades as phases of slow and fast eye movements, respectively. Firstly, the raw 1 kHz *x* and *y* gaze signals were smoothed by applying a Savitzky-Golay filter. We then applied an adaptive velocity threshold algorithm to this smoothed signal, thereby obtaining candidate fixation phases, with everything in between being candidate saccade phases. Thereafter, we applied two basic merging criteria. Firstly, saccade candidates with amplitude $< 1.0°$ were removed, thereby merging neighbouring fixation candidates. Subsequently, all fixation candidates with duration $< 60$ms were removed. This procedure successfully removes large differences in oculomotor event classification between different algorithms [28]. Gaze amplitudes and velocities in pixels were converted to degrees of visual angle by multiplying their values by 0.0186 [23]. Lastly, blinks were detected by finding periods in which no pupil data was measured. All events which lasted less than 30ms, or more than 3 seconds, were removed. These thresholds were set so that neither brief nor longer periods of data loss would be incorrectly detected as blinks.

### Data pre-processing

After extraction of oculomotor metrics, data of each participant was split into 240 chunks of 30 seconds each. However, the last chunk was often shorter than 30 seconds, and some chunks had too much data loss. As such, these chunks were discarded, resulting in 3327 data points. Then, heart rate detection was performed over the raw pulse oximetry signal within these chunks, using HeartPy [29]. Thirty seconds were selected as chunk size because it provides a balance between sufficient data per chunk ($> 20$ fixations on average, and sufficient time for accurate heart rate detection), and a sufficient number of chunks for machine learning purposes.

For each chunk, twelve features were extracted: (1, 2, 3) the *duration* of each fixation, saccade, and blink event; (4, 5) the *amplitude* of each fixation and saccade event; (6, 7) the *peak velocity* of each fixation and saccade event; (8, 9) the *mean velocity* of each fixation and saccade event; and (10, 11, 12) the *count* of fixation, saccade, and blink events in that chunk.

We took a two-fold approach to testing whether oculomotor metrics can be sufficient predictors of heart rate. Firstly, we posed that the prediction of heart rate could be considered a regression problem, in which we aimed to predict heart rate on a continuous scale. Secondly, we posed that the prediction of heart rate could also be considered a binary classification problem (above some threshold or below some threshold). This approach can be useful when the aim is to only predict whether someone is either excessively or insufficiently aroused.

To prepare our dependent variable for binary classification, the heart rate of each chunk was expressed as a *z*-score; the number of standard deviations from the median heart rate of that respective participant over the full film. Each *z*-score was then converted to a binary variable–namely *low* if $z < -.5$, and *high* if $z > .5$. All other chunks were considered *neutral* and discarded. Since the distributions of heart rate were often skewed, and due to slightly differing

amounts of data loss, our binarization did not result in equally large samples of *high* and *low* labels. As a result, 513 chunks were below the threshold, and 607 chunks were above the threshold. A total of 1120 data points remained after binarizing the heart rate data. Distributions of each feature, split per label, are reported in Fig 1.

### Feature pre-processing

As is common in machine learning pipelines, our classifier required an equally long set of features per chunk of data, and the described feature set did not comply with this requirement. For example, if 30 saccades were made within one chunk, and 40 saccades were made in another chunk, the *peak saccade velocity* variable would contain 30 and 40 values for each of those chunks, respectively. Therefore, our data needed to be aggregated. Three methods were explored, as outlined in the next subsections.

**Averaging.** Within each chunk, the average of each of the twelve features was computed, providing one value per feature for each chunk. This approach provides the most intuitive insight into the amount of information contained within each feature, which in turn contributes towards correct classification.

**Feature explosion.** It could be argued that simply calculating the mean value over features would discard relevant information, since, for instance, the mean saccade velocity across chunks may be equal, but the variance across chunks could be different. Similar to the approach of Kootstra et al. (2020), a set of 13 statistical descriptors (e.g., *mean*, *variance*, *uniformity*) was employed to describe the distribution of each of the features 1–9 within each chunk (see S1 Table for a full list of the statistical descriptors). Through this method, the dataset was thus 'exploded' and contained *3 count features + (9 features × 13 descriptors)* = 119 features.

**Feature explosion and dimensionality reduction.** To aid interpretation of these 119 features, each of the oculomotor metrics was to be described in at most two variables. To this end, each of the nine exploded features was reduced from a description of dimensionality 13 to a description of dimensionality 2 by taking the two components with the highest explained variance from Principal Component Analysis (PCA). This resulted in a set of *3 count features + (9 features × 2 descriptors)* = 21 features. On average, the first two components taken from PCA provided an explained variance of 98.98% for the nine features.

### Regression pipeline

We fitted a multiple linear regression with heart rate per chunk as the dependent variable, and either of the features obtained by the methods outlined above as independent variables. In addition, a similar but polynomial regression was fitted, to identify possible non-linear links. All regression models were fit to the train set and $R^2$ was evaluated on the test set.

### Machine learning pipeline

Logistic Regression, K-Nearest Neighbours and Random Forest Classifier were used to predict high versus low heart rate from oculomotor metrics. Each type of model was run independently 50 times, with a new 80/20% stratified train/test split for each run, and with the default set of parameters as provided by scikit-learn. On average over those 50 runs, and across the three different pre-processing approaches, the Random Forest classifier performed best of the three models, and thus this model was selected for further optimization (see Table 2).

Subsequently, hyperparameter optimization of the Random Forest classifier was implemented over the *number of trees* (range 10–200; step size 1) and the *maximum depth per tree* (range 1–30 + unlimited depth). All other hyperparameters were kept as default. We then constructed 500 candidate combinations of hyperparameters by randomly sampling from their

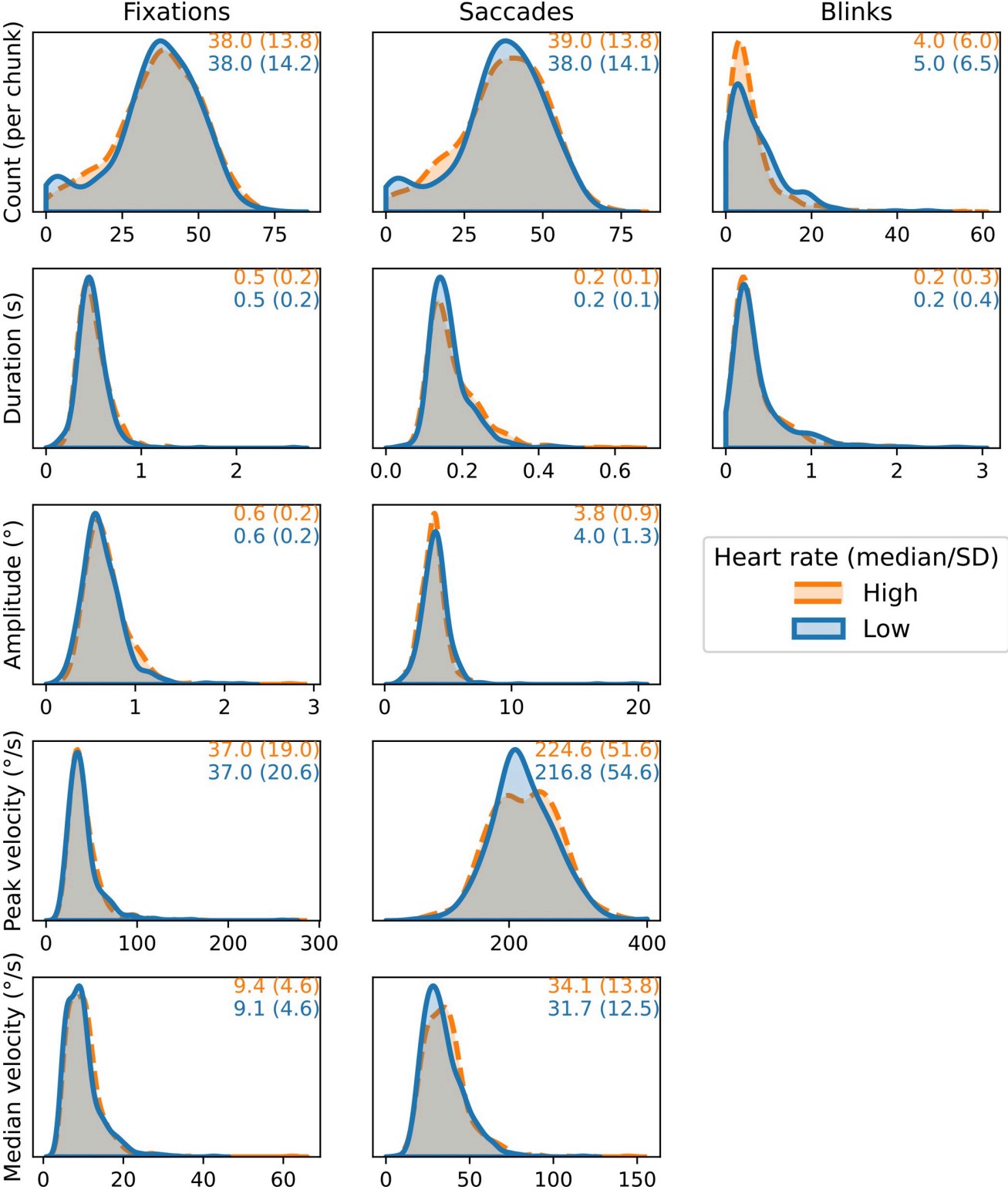

**Fig 1. Distributions (kernel density estimation) for each of the twelve features, per label (*high* or *low* heart rate).** The distributions are computed over all chunks for all participants, thus 1120 data points per feature (513 low, 607 high). Orange and blue values indicate median and standard deviation of each of the high and low heart rate distributions, respectively.

**Table 1. Outcomes ($R^2$) of the linear regression models, per pre-processing approach.** Features were either derived directly from the pre-processing approach, or with added second-degree polynomials for each feature. Models were fit to the 80% train set and evaluated on the 20% test set.

| | $R^2$ | $R^2$ (with 2nd degree polynomials) |
|---|---|---|
| Averaging | .18 | .30 |
| Explosion | .21 | $<. 01$ |
| Explosion + reduction | .17 | .12 |

specified distributions. Each candidate combination was assigned the same 80% training set and was evaluated on that set using 5-fold stratified cross-validation and Area Under the Curve (AUC) as performance metric. An AUC of 0.50 constitutes classification at chance level and 1.0 constitutes complete accuracy. The model and parameter combination that led to the best cross-validation result was then tested on the 20% holdout set. To compensate for randomness effects in the sampling of the training- and test sets, and in the sampling of hyperparameters, this search process was repeated 50 times and means and standard deviations are reported.

Finally, the contributions of all features towards correct classification were extracted from the best-performing model using permutation importance [30]. For each feature, a one-sample *t*-test was performed to test whether that feature's importance differed significantly from the overall mean (higher importance is better; *t*-test α = .05).

## Results

### Regression

$R^2$ for regression models ranged between $< .01$ and .30 (see Table 1 for full results), indicating that oculomotor metrics provide limited information towards prediction of heart rate as a continuous variable.

### Classification

Overall, the *averaging* pre-processing approach provided the best performance at classifying whether a participant had a high- versus low heart rate within a chunk (AUC = .696). The model pre-selection results and the results of optimization are reported in Table 2.

The best-performing model performed consistently above chance and achieved an average AUC of .703 (*SD = .02*) on the cross-validation sets, and an average AUC of .698 (*SD = .04*) on the test sets over 50 independent runs. An overview of the best models and the runner-up models is reported in Table 3.

The extraction of feature importance's revealed blink rate, duration, and features associated with oculomotor movement to be most predictive of heart rate ([fixation and saccadic] median

**Table 2. AUCs of the model pre-selection process (averaged over 50 independent model runs).**

| | Logistic Regression | K-Nearest Neighbours | Random Forest | Random Forest + optimization[a] |
|---|---|---|---|---|
| Averaging | .622 | .617 | **.696** | **.698** *(.04)* |
| Explosion | .590 | .588 | .660 | .664 *(.05)* |
| Explosion + reduction | .614 | .585 | .666 | .678 *(.04)* |

[a]The average *(SD)* outcome on the test set over 50 runs of the optimized model is reported.

**Table 3. AUCs and parameters of the best-performing models and runner-up models resulting from hyperparameter search (on the *averaging* pre-processing approach).**

|  | Model rank 1 | Model rank 2 | Model rank 3 |
|---|---|---|---|
| Cross-validation performance (AUC) | .703 | .701 | .700 |
| 20% holdout set performance (AUC) | .698 | - | - |
| Average number of trees | 126.5 | 135.0 | 127.2 |
| Average maximum depth per tree | 20.2[a] | 19.3[a] | 19.5[a] |

Model ranks were defined based on cross-validated classification performance. All values are averages over 50 runs. In each run, only the best model was tested against the test set.

[a]Includes at least one model where the maximum depth was unlimited

velocity, saccadic peak velocity; Fig 2). All other features were found to contribute worse-than-average towards classification.

## Discussion

In the current study, we investigated how well oculomotor metrics may predict heart rate and which of these features drive this prediction predominantly. To this end, we used a public dataset of participants whose physiological data were obtained while watching the 1994 *Forrest Gump* motion picture. Although oculomotor metrics provided limited predictive value for linear and polynomial regressions (up to $R^2$ of .30), a Random Forest model could predict high-versus low heart rate consistently at above-chance level. In this model, the features which

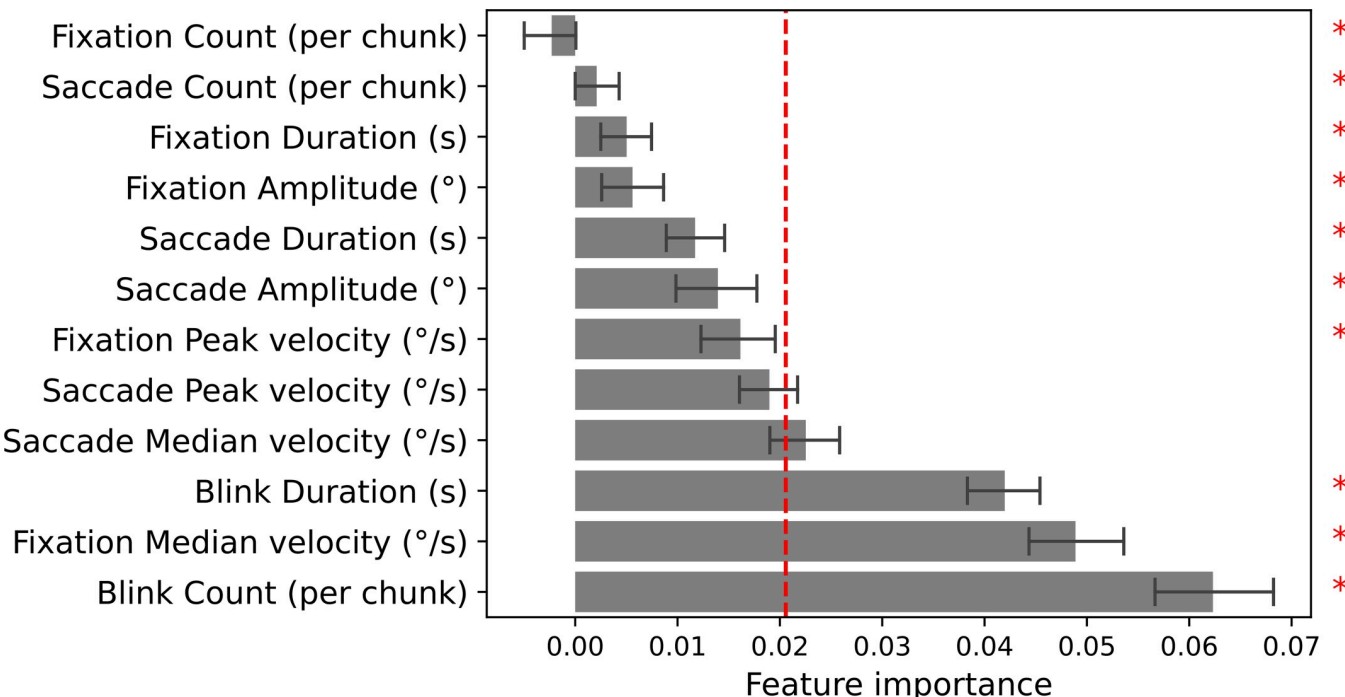

**Fig 2. Mean (± 95% CI) feature importance's as extracted from the best-performing model of each of the 50 runs.** Higher values imply a higher degree of information within the variable. The vertical dashed line represents the overall mean of all importance values. The asterisks represent where feature importance's differed significantly from the overall mean.

contributed most strongly towards correct classification were blink rate, blink duration, and the median velocity within fixations and saccades, and the saccadic peak velocity.

Interestingly, each of the features that contributed most strongly pertains to either information regarding blinks, or regarding oculomotor movement (velocities and amplitudes), and not so much to durations or counts of fixations and saccades. The importance of *blink rate* and *blink duration* provides support for the suggested link between an altered rate of eyeblinks and changes in arousal [11, 12] and changes in heart rate metrics [13]. At first sight, the relative importance of *fixation velocity* might be surprising, since fixations are spatially stable. However, differences in fixation velocities may be the result of physiological drift or microsaccades, sometimes referred to as fixational drift or fixational eye movements [31]. The occurrence of microsaccades has been found to be positively coupled to heartbeat, and may thus explain the amount of information captured in the *fixation velocity* variable [32]. The *peak* and *median velocity of saccades* are fourth and fifth in the list of informative features, which aligns with earlier literature which suggested that saccadic peak velocity indicates mental effort [7, 8] and motivation [9]–two cognitive processes closely linked to modulations in arousal.

Feature importance, however, does not indicate specifically which aspect of a distribution provides the most information towards correct classification. This makes it difficult to speculate about the direction of the effect of the included features, further complicated by inconsistencies in the literature. For instance, microsaccades occur more frequently with high mental effort in some tasks, but not in others [33, 34], suggesting that the modulation of eye movement and heart rate by the arousal system is highly task-dependent. This is further evidenced by the fact that we find increased saccadic- and fixational velocities in high heart rate periods, whereas it is usually found that saccadic and fixational velocity are negatively correlated with arousal [7, 33]. While, except within velocity, no consistently different medians within features were found between low- and high heart rate periods, it is remarkable that standard deviations were consistently equal or higher when heart rate was low, as compared to when it was high (with the exception of median saccade velocity). High arousal levels could be associated with a reduction in variability in oculomotor behaviour, as is the case with heart rate [35].

Based on these findings, we speculate that heart rate is not only linked to fixational eye movements [32], but to oculomotor movements in general. This link might come into place due to changes in the common underlying arousal system, or merely as an effect of changes in blood pressure during the heartbeat cycle. Our findings therefore suggest that a substantial portion of oculomotor behaviour is linked to heart rate, and not only by top-down goals of the beholder [22], or bottom-up visual features of the scene [36], as is commonly assumed. To this end, other physiological indicators could be compared to oculomotor metrics in their ability to predict heart rate. Because there is no unified definition of arousal, investigating the links between the aforementioned indicators would allow to isolate more specific subcomponents of arousal, and improve our definition of the term.

Speculating about neural underpinnings for a link between the oculomotor features described here and heart rate, we see a potential role for the locus coeruleus, a sympathetic center in the brain that acts antagonistically to parasympathetic activation associated with heart rate variability [37]. The noradrenergic locus coeruleus affects oculomotor behavior mainly via its inputs to the superior colliculus that is crucial in bringing about several oculomotor behaviours [38]. Note that locus coeruleus-centered and superior colliculus-centered circuits have been associated with differential attentional functions at the level of the brain stem, including alerting and orienting [38].

Another putative candidate might be the hypothalamus [13] (though bidirectionally linked to the locus coeruleus [38]) which modulates activity in the autonomous nervous system. Its link to the basal ganglia (and changes in the dopamine system) might explain the relation

between blinks and heart rate, as changes in dopamine levels in the basal ganglia are monitored with changes in blink rate [13]. Although a relation between heart rate and oculomotor features and these two brain regions seems plausible, it is important to stress that this currently mere speculation and should be the subject of future research.

The current study is limited in its comprehensiveness of oculomotor features. For instance, pupil dilation has been shown to encode aspects of arousal [5, 6]. However, these measurements are distorted when gaze position changes [39] and could therefore not be reliably measured. Another step could be to link eye movements to on-screen movements in order to obtain smooth pursuits. This might be meaningful, as deviations in smooth pursuit trajectories have been found to be indicative of mental effort–and thus by proxy arousal [40]. Currently, smooth pursuits are likely to be captured within fixations and saccades at the high- and low ends of their respective velocity distributions. Lastly, as a next step, microsaccades could be investigated in detail [41, 42]. This would require robust detection algorithms that work without static scenes and with monocular data.

Different parameter sets and pre-processing approaches all lead to similar model performances, as shown in Tables 1–3. For example, the pre-processing approaches of averaging and feature explosion without reduction led to very similar outcomes in classification accuracy. However, the averaging approach required less processing time and can be interpreted more intuitively. Furthermore, classification accuracy could be improved by using more complex models, but again at the cost of interpretability.

The current study does not directly provide a method for the real-time prediction of heart rate from oculomotor metrics, since the proposed Random Forest classification pipeline requires that a baseline heart rate is established from which to derive *low* or *high* heart rate labels as the dependent variable. However, future research may attempt to establish a baseline heart rate measurement before the start of a given task and subsequently investigate whether the prediction of heart rate can be conducted in real-time.

## Conclusion

In conclusion, oculomotor metrics obtained during a naturalistic viewing task contain sufficient information to predict *high* versus *low* heart rates above chance during that same task. These findings not only establish oculomotor metrics as unobtrusively measurable predictors of heart rate, but open new pathways for investigation of the link between oculomotor metrics and various indicators of arousal.

## Supporting information

**S1 Table. List of statistical descriptors.**
(DOCX)

## Acknowledgments

The authors would like to thank Roy Hessels for his input regarding fixation detection.

## Author Contributions

**Conceptualization:** Alex J. Hoogerbrugge, Christoph Strauch, Zoril A. Oláh, Edwin S. Dalmaijer, Stefan Van der Stigchel.

**Formal analysis:** Alex J. Hoogerbrugge, Christoph Strauch.

**Funding acquisition:** Stefan Van der Stigchel.

**Supervision:** Christoph Strauch, Tanja C. W. Nijboer, Stefan Van der Stigchel.

**Validation:** Zoril A. Oláh, Edwin S. Dalmaijer, Tanja C. W. Nijboer, Stefan Van der Stigchel.

**Visualization:** Alex J. Hoogerbrugge.

**Writing – original draft:** Alex J. Hoogerbrugge, Christoph Strauch.

**Writing – review & editing:** Alex J. Hoogerbrugge, Christoph Strauch, Zoril A. Oláh, Edwin S. Dalmaijer, Tanja C. W. Nijboer, Stefan Van der Stigchel.

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
