## [Decision Letter · Decision Letter 0]

23 May 2022

PONE-D-22-08472Seeing the Forrest through the trees: Oculomotor metrics are linked to heart ratePLOS ONE

Dear Dr. Hoogerbrugge,

Thank you for submitting your manuscript to PLOS ONE. After careful consideration, we feel that it has merit but does not fully meet PLOS ONE’s publication criteria as it currently stands. Therefore, we invite you to submit a revised version of the manuscript that addresses the points raised during the review process.

We look forward to receiving your revised manuscript.

Kind regards,

Enkelejda Kasneci, Ph.D.

Academic Editor

PLOS ONE

Journal Requirements:

“This work was supported by ERC [ERC-CoG-863732].”

“This work was supported by ERC [ERC-CoG-863732], https://erc.europa.eu/, awarded to SVdS. The funders had no role in study design, data collection and analysis, decision to publish, or preparation of the manuscript.”

Reviewers' comments:

Reviewer's Responses to Questions

**Comments to the Author**

1. Is the manuscript technically sound, and do the data support the conclusions?

Reviewer #1: Yes

Reviewer #2: Yes

2. Has the statistical analysis been performed appropriately and rigorously? 

Reviewer #1: Yes

Reviewer #2: Yes

3. Have the authors made all data underlying the findings in their manuscript fully available?

Reviewer #1: Yes

Reviewer #2: Yes

4. Is the manuscript presented in an intelligible fashion and written in standard English?

Reviewer #1: Yes

Reviewer #2: Yes

5. Review Comments to the Author

Reviewer #1: The authors investigated the connection between faxation and saccade metric and changes in heart rate during film viewing. Based on the gaze data, they trained random forest models (both for regression and classification) in order to predict participants‘ heart rate during specific segments oft he movie.

There are a lot of aspects of this study that I like. They use a public dataset and provide their analysis scripts, so reproducability should be possible. I also think that the topic of oculomotor events and physiological arousal is very interesting and rarely investigated. I commend the authors for the work they have put into this manuscript!

The study design an methodology seem fine for the most part and the language is easy to understand. However, there a several concerns that need to be addressed before I can fully approve oft he manuscript as it is presented here.

The major concern is potentially confounding factors that are not looked into: namely blinks and smooth pursuits. Blinks are disregarded even though they could impact eye movements or may even add an additional puzzle piece fort he link that the authors are investigating. Nakano and Kuriyama (2017) showed that spontaneous blinks (that often occur at attentional breakpoints, which would be very fitting fort he movie context) are associated with increased heart rate, so this avenue seems like a good addition tot he feature portfolio employed by the authors. Blinks could also distort gaze metrics if they occur to frequently, so at least stating how they are addressed would improve my confidence in the presented results.

Smooth pursuits are likely relevant for gaze analysis of a movie as slowly moving targets or cameras moving in relation to actors or objects occur rather often. Smooth pursuits could especially skew fixation metrics like dispersion, velocity, or amplitude if not treated accordingly. Similar tot he issue with blinks, this should be addressed in some way.

Finally, in their discussion the authors mention microsaccades as a potential explanation for their high importance of fixation-related metrics. I would like to see this thought explored in more detail! The sampling frequency oft he eye tracker is 1000Hz, so reliably investigating microsaccades should be possible. This could really help to shed more light on the link between eye movements and heart rate. For datasets with lower sampling frequencies where microsaccades cannot be determined reliably, the authors‘ currently suggested metrics would then work as a substitute.

Minor issues:

- In the context of pupil diameter, the authors state that „… among eye tracking scientists there is no unified concept of how fixations and saccades should be defined – and thus the application of differing fixation- and saccade detection techniques may result in differing outcomes, even if they are applied to the same dataset (Hessels et al., 2018).“ The same argument likely applies to the metrics used by the authors as velocity and aplitude of fixations and saccades are a major feature in their approach.

- It may be tough to disentagle eye movement characteristics that are caused by physiology and arousal from those that appear stimulus driven which opens the door for many confounding factors.

- The imbalance on the two classes is not addressed, eventhough it is only slightly out of balance.

- How much variance was preserved by the PCA? This may help to judge the feature explosion and reduction approach.

- Was the z-normalization as a preprocessing step for forming two classes performed on a participant level or globally with the dataset as a whole?

- The figures do not seem to scale well. The authors may need to redo them as vectored graphics to help with readability.

Reviewer #2: In this study, the authors took data from 14 participants viewing films and compared oculomotor metrics to heart rate, querying whether they would be linked in a way where noninvasive oculomotor monitoring could predict heart rate.

In terms of analytics, the authors found that heart rate had to be split into high vs low, rather than as a continuous variable. This limits the predictiveness of the oculomotor metrics, as noted clearly by the authors. They found that 4 metrics: fixational and saccadic velocities, saccade peak velocity, and saccade amplitude were the best features for a random forest model to categorize each chunk of the movie watching as high or low heart rate better than chance.

This is a simple and elegant study. It is an initial proof of concept study (my description rather than the authors), towards the stated goal of using oculomotor metrics to predict heart rate in real time. The authors clearly note that the current method cannot be used in real time due to needing a baseline heart rate for the task at hand, but future work could improve classification accuracy or determine if a pre-task baseline can be used for real time prediction.

My concerns with the manuscript are based upon the short discussion. There are a couple of areas where the discussion could put the results in more context for the benefit of the readers.

The manuscript discusses the 4 features as feeding into the random forest model, and then some interpretations about why for each feature. However, the discussion does not clearly state the differences in light of low/high heart rate. For example, fixation velocity is appropriately described as potentially reflecting microsaccades, where microsaccade rate can vary by arousal or complexity, but which way? Alertness can improve fixational stability when focused on a difficult task, but arousal can increase exploratory gaze behavior. How is the metric of fixation velocity related to low vs high heart rate chunks, in this task of movie watching? The same lack of explanation occurs for the other 3 featured metrics. Or is it a given pattern/combination? While each metric's distribution is depicted in Figure 1, qualitatively there's a more visible difference between high/low heart rates for counts than for median velocities, yet the analytics showed median velocities over counts.

It would be useful for information about how the metrics (as a pattern, or individually) are related to the two heart rate categories to better relate the oculomotor system to heart rate, as movie watching and heart rate is related to limbic responses rather than the references in the discussion relating arousal to task difficulty and other achievement-style contexts. As the authors note, it may be due to a common underlying process. Underlying limbic system mechanisms could have a different effect on the oculomotor system than from say ascending reticular activating system or prefrontal-mediated executive functions such as attentional and inhibitory control.

And that is a second area to potentially added to the discussion - if there's a common mechanism, what would that putative mechanism be? Any known connectivity to the oculomotor system? Are these differences arising from oculomotor nuclei in the brainstem, subcortical areas, prefrontal? The results are clear, but the implications or interpretations that could link them more broadly are missing.

6. PLOS authors have the option to publish the peer review history of their article (what does this mean?). If published, this will include your full peer review and any attached files.

Reviewer #1: **Yes: **Tobias Appel

Reviewer #2: No

---

## [Author Response · Author response to Decision Letter 0]

1 Jul 2022

We kindly thank both of the reviewers for their positive and useful feedback, and hope that the implemented changes and responses address all points adequately. As part of the revision, we have reanalyzed the data and completely revised the results, along with both figures and all tables.

Reviewer #1 

The authors investigated the connection between faxation and saccade metric and changes in heart rate during film viewing. Based on the gaze data, they trained random forest models (both for regression and classification) in order to predict participants‘ heart rate during specific segments oft he movie.

There are a lot of aspects of this study that I like. They use a public dataset and provide their analysis scripts, so reproducability should be possible. I also think that the topic of oculomotor events and physiological arousal is very interesting and rarely investigated. I commend the authors for the work they have put into this manuscript!

The study design an methodology seem fine for the most part and the language is easy to understand. However, there a several concerns that need to be addressed before I can fully approve oft he manuscript as it is presented here.

1. The major concern is potentially confounding factors that are not looked into: namely blinks and smooth pursuits. Blinks are disregarded even though they could impact eye movements or may even add an additional puzzle piece fort he link that the authors are investigating. Nakano and Kuriyama (2017) showed that spontaneous blinks (that often occur at attentional breakpoints, which would be very fitting fort he movie context) are associated with increased heart rate, so this avenue seems like a good addition tot he feature portfolio employed by the authors. Blinks could also distort gaze metrics if they occur to frequently, so at least stating how they are addressed would improve my confidence in the presented results.

Thank you for this excellent remark. We have now included blink rates and blink duration in our detection algorithm, and as features in our models. 

Whereas our best model previously achieved an AUC of .646 on the test set, it now classifies with an AUC of .698, meaning that blinks indeed add meaningful information to our model.

We have addressed this in various places in the manuscript, including completely revising Figures 1 and 2, Tables 1, 2, 3 and the results section. The most notable changes are as follows:

[lines 53-56] “Additionally, it has been shown that changes in arousal are paired with an altered rate of eyeblinks (e.g., Maffei & Angrilli, 2019; Wood & Hassett, 1983), and that spontaneous eyeblinks occur in tandem with an increase in heart rate variability (Nakano & Kuriyama, 2017).”

[lines 117-120] “Lastly, blinks were detected by finding periods in which no pupil data was measured. All events which lasted less than 30ms, or more than 3 seconds, were removed. These thresholds were set so that neither brief nor longer periods of data loss would be incorrectly detected as blinks.”

[lines 249-252] “The importance of blink rate and blink duration provides support for the suggested link between an altered rate of eyeblinks and changes in arousal (Maffei & Angrilli, 2019; Wood & Hassett, 1983) and changes in heart rate metrics (Nakano & Kuriyama, 2017).”

2. Smooth pursuits are likely relevant for gaze analysis of a movie as slowly moving targets or cameras moving in relation to actors or objects occur rather often. Smooth pursuits could especially skew fixation metrics like dispersion, velocity, or amplitude if not treated accordingly. Similar tot he issue with blinks, this should be addressed in some way.

We agree that smooth pursuits could be a relevant metric to include. However, we have found smooth pursuits to be difficult to reliably extract as an independent metric from this dataset. Namely, we would need to know whether there are pursuable objects on the screen, and where they are, at any given moment – information that is currently not available in this public data set. 

A second option is to set a filter (e.g., fixational events with velocity > 2 degrees per second) and label those events as smooth pursuits. Smooth pursuit events are currently likely to be captured within both fixations and saccades at the high- and low ends of the velocity distributions respectively. Including smooth pursuits like this would therefore be a relatively coarse approach that will likely not only include smooth pursuits, but also other types of oculomotor behavior. To ensure interpretability we thus prefer to not include smooth pursuits as a separate feature in the final analysis. Yet, we fully agree with the reviewer here and discuss the potential of smooth pursuits as further predictor as follows: 

[lines 307-312] “Another step could be to link eye movements to on-screen movements in order to obtain smooth pursuits. This might be meaningful, as deviations in smooth pursuit trajectories have been found to be indicative of mental effort – and thus by proxy arousal (Kosch et al., 2018). Currently, smooth pursuits are likely to be captured within fixations and saccades at the high- and low ends of their respective velocity distributions.”

3. Finally, in their discussion the authors mention microsaccades as a potential explanation for their high importance of fixation-related metrics. I would like to see this thought explored in more detail! The sampling frequency of the eye tracker is 1000Hz, so reliably investigating microsaccades should be possible. This could really help to shed more light on the link between eye movements and heart rate. For datasets with lower sampling frequencies where microsaccades cannot be determined reliably, the authors‘ currently suggested metrics would then work as a substitute.

Microsaccades are certainly informative (particularly regarding the link to heart rate), but are hardly retrievable during free viewing with sufficient confidence. We agree that sampling rate and accuracy of the tracker would be sufficient to track microsaccades – but still, a reliable estimation of microsaccades is questionable (see explanation above). For one, Engbert & Kliegl (2003) describe microsaccade detection for binocular data, and while it is also possible with monocular data (like in the dataset used here), this is more prone to noise. Furthermore, classical studies using microsaccades strictly ensure fixation, which is impossible to do during free viewing and would almost certainly drive-up noise even further. We now address this interesting further step by discussing: 

[lines 312-314] “Lastly, as a next step, microsaccades could be investigated in detail (Duchowski et al., 2020; Engbert & Kliegl, 2003). This would require robust detection algorithms that work without static scenes and with monocular data.” 

4. Minor issues:

a. In the context of pupil diameter, the authors state that „… among eye tracking scientists there is no unified concept of how fixations and saccades should be defined – and thus the application of differing fixation- and saccade detection techniques may result in differing outcomes, even if they are applied to the same dataset (Hessels et al., 2018).“ The same argument likely applies to the metrics used by the authors as velocity and aplitude of fixations and saccades are a major feature in their approach.

We agree with the reviewer that this is an important issue and now addressed it more clearly. Namely, we applied merging criteria which have been demonstrated to reduce such event-detection induced effects to a minimum. We now write:

[lines 111-115] “Thereafter, we applied two basic merging criteria. Firstly, saccade candidates with amplitude < 1.0° were removed, thereby merging neighbouring fixation candidates. Subsequently, all fixation candidates with duration < 60ms were removed. This procedure successfully removes large differences in oculomotor event classification between different algorithms (Hooge et al., 2022).”

Furthermore, our main intention in the quoted paragraph was to emphasize that including additional features – or a combination of several features – would eventually make models more robust to this variance. We have now accentuated this in the manuscript:

[lines 64-68] “As such, incorporating several metrics which can be independently extracted (e.g., pupil size, oculomotor movement, blinks) would improve robustness of the model, as it reduces dependence on one single extraction technique. This also applies to cases in which pupil dilation measurements are unreliable or missing, or the eye tracker’s sampling rate is too low to extract peak saccade velocities.”

b. It may be tough to disentagle eye movement characteristics that are caused by physiology and arousal from those that appear stimulus driven which opens the door for many confounding factors.

We fully agree with the reviewer here – this endeavour will be an ongoing one for the field to which we try to contribute a first step. We see the combination of several indicators of arousal (here heart rate) with oculomotor metrics as a promising step in this direction, yet these results only yield correlational not causal evidence. We address this by writing in the discussion: 

[lines 280-284] “Our findings therefore suggest that a substantial portion of oculomotor behaviour is linked to heart rate, and not only by top-down goals of the beholder (e.g., Kootstra et al., 2020), or bottom-up visual features of the scene (e.g., Itti & Koch, 2000), as is commonly assumed. To this end, other physiological indicators could be compared to oculomotor metrics in their ability to predict heart rate.” 

c. The imbalance on the two classes is not addressed, eventhough it is only slightly out of balance.

We have clarified this in the manuscript:

[lines 144-147] “All other chunks were considered neutral and discarded. Since the distributions of heart rate were often skewed, and due to slightly differing amounts of data loss, our binarization did not result in equally large samples of high and low labels.”

d. How much variance was preserved by the PCA? This may help to judge the feature explosion and reduction approach.

Taking the first two components from PCA for each of the features provided an overall average of 98.98% explained variance. We here report the full table, but only include the overall average in the manuscript for brevity.

Movement type Feature Explained variance (%)

Blink Duration 91,93

Fixation Amplitude 99,88

Fixation Duration 99,55

Fixation Median velocity > 99.9

Fixation Peak velocity > 99.9

Saccade Amplitude > 99.9

Saccade Duration 99,43

Saccade Median velocity > 99.9

Saccade Peak velocity > 99.9

Overall 98,98%

Adjusted in the manuscript:

[lines 181-183] “On average, the first two components taken from PCA provided an explained variance of 98.98% for the nine features.”

e. Was the z-normalization as a preprocessing step for forming two classes performed on a participant level or globally with the dataset as a whole?

z-normalization of heart rate was performed per participant. We now more clearly state:

[lines 141-143] “To prepare our dependent variable for binary classification, the heart rate of each chunk was expressed as a z-score; the number of standard deviations from the median heart rate of that respective participant over the full film.”

f. The figures do not seem to scale well. The authors may need to redo them as vectored graphics to help with readability.

Thank you for noticing this. We have changed the figure format.

Reviewer #2 

In this study, the authors took data from 14 participants viewing films and compared oculomotor metrics to heart rate, querying whether they would be linked in a way where noninvasive oculomotor monitoring could predict heart rate.

In terms of analytics, the authors found that heart rate had to be split into high vs low, rather than as a continuous variable. This limits the predictiveness of the oculomotor metrics, as noted clearly by the authors. They found that 4 metrics: fixational and saccadic velocities, saccade peak velocity, and saccade amplitude were the best features for a random forest model to categorize each chunk of the movie watching as high or low heart rate better than chance.

This is a simple and elegant study. It is an initial proof of concept study (my description rather than the authors), towards the stated goal of using oculomotor metrics to predict heart rate in real time. The authors clearly note that the current method cannot be used in real time due to needing a baseline heart rate for the task at hand, but future work could improve classification accuracy or determine if a pre-task baseline can be used for real time prediction.

My concerns with the manuscript are based upon the short discussion. There are a couple of areas where the discussion could put the results in more context for the benefit of the readers.

1. The manuscript discusses the 4 features as feeding into the random forest model, and then some interpretations about why for each feature. However, the discussion does not clearly state the differences in light of low/high heart rate. For example, fixation velocity is appropriately described as potentially reflecting microsaccades, where microsaccade rate can vary by arousal or complexity, but which way? Alertness can improve fixational stability when focused on a difficult task, but arousal can increase exploratory gaze behavior. How is the metric of fixation velocity related to low vs high heart rate chunks, in this task of movie watching? The same lack of explanation occurs for the other 3 featured metrics. Or is it a given pattern/combination? While each metric's distribution is depicted in Figure 1, qualitatively there's a more visible difference between high/low heart rates for counts than for median velocities, yet the analytics showed median velocities over counts.

Thank you for raising this important theoretical question. Indeed, it is useful to know how specifically any of our given features link to heart rate. Of course, a combination of features might explain more, by interactive effects, than individual features. We now explain why feature importances contain limited information about directionality, and that these do not allow conclusions about patterns within features. We relate our findings to inconsistencies within literature, as pointed out by the reviewer. We have also included medians and standard deviations to the distributions in Figure 1. As can be seen, medians are often very similar across high- and low heart rates, regardless of whether these features contribute strongly to the models. Yet, standard deviations are consistently equal or higher when heart rate is low, compared to when it is high (with the exception of median saccade velocity), which indicates that high arousal levels could be associated with reduced variability in oculomotor metrics.

We now address this by writing:

[lines 262-276]: “Feature importance, however, does not indicate specifically which aspect of a distribution provides the most information towards correct classification. This makes it difficult to speculate about the direction of the effect of the included features, further complicated by inconsistencies in the literature. For instance, microsaccades occur more frequently with high mental effort in some tasks, but not in others (Pastukhov & Braun, 2010; Siegenthaler et al., 2014), suggesting that the modulation of eye movement and heart rate by the arousal system is highly task-dependent. This is further evidenced by the fact that we find increased saccadic- and fixational velocities in high heart rate periods, whereas it is usually found that saccadic and fixational velocity are negatively correlated with arousal (Di Stasi et al., 2013; Siegenthaler et al., 2014). While, except within velocity, no consistently different medians within features were found between low- and high heart rate periods, it is remarkable that standard deviations were consistently equal or higher when heart rate was low, as compared to when it was high (with the exception of median saccade velocity). High arousal levels could be associated with a reduction in variability in oculomotor behaviour, as is the case with heart rate (Kazmi et al., 2016).”

2. It would be useful for information about how the metrics (as a pattern, or individually) are related to the two heart rate categories to better relate the oculomotor system to heart rate, as movie watching and heart rate is related to limbic responses rather than the references in the discussion relating arousal to task difficulty and other achievement-style contexts. As the authors note, it may be due to a common underlying process. Underlying limbic system mechanisms could have a different effect on the oculomotor system than from say ascending reticular activating system or prefrontal-mediated executive functions such as attentional and inhibitory control.

And that is a second area to potentially added to the discussion - if there's a common mechanism, what would that putative mechanism be? Any known connectivity to the oculomotor system? Are these differences arising from oculomotor nuclei in the brainstem, subcortical areas, prefrontal? The results are clear, but the implications or interpretations that could link them more broadly are missing.

These are interesting questions and agree that a discussion on putative mechanisms is interesting. We believe that hypothalamus and locus coeruleus are most likely underlying centres, yet this is not necessarily exclusive and cannot be answered exhaustively with the current data. We now mention the following in the discussion:

[lines 288-303] “Speculating about neural underpinnings for a link between the oculomotor features described here and heart rate, we see a potential role for the locus coeruleus, a sympathetic center in the brain that acts antagonistically to parasympathetic activation associated with heart rate variability (Mather et al., 2017). The noradrenergic locus coeruleus affects oculomotor behavior mainly via its inputs to the superior colliculus that is crucial in bringing about several oculomotor behaviours (Strauch et al., 2022). Note that locus coeruleus-centered and superior colliculus-centered circuits have been associated with differential attentional functions at the level of the brain stem, including alerting and orienting (Strauch et al., 2022). Another putative candidate might be the hypothalamus (Nakano & Kuriyama, 2017; though bidirectionally linked to the locus coeruleus, Strauch et al., 2022) which modulates activity in the autonomous nervous system. Its link to the basal ganglia (and changes in the dopamine system) might explain the relation between blinks and heart rate, as changes in dopamine levels in the basal ganglia are monitored with changes in blink rate (Nakano & Kuriyama, 2017). Although a relation between heart rate and oculomotor features and these two brain regions seem plausible, it is important to stress that this currently mere speculation and should be the subject of future research.”

---

## [Decision Letter · Decision Letter 1]

19 Jul 2022

Seeing the Forrest through the trees: Oculomotor metrics are linked to heart rate

PONE-D-22-08472R1

Dear Dr. Hoogerbrugge,

We’re pleased to inform you that your manuscript has been judged scientifically suitable for publication and will be formally accepted for publication once it meets all outstanding technical requirements.

Kind regards,

Enkelejda Kasneci, Ph.D.

Academic Editor

PLOS ONE

Reviewers' comments:

Reviewer's Responses to Questions

**Comments to the Author**

1. If the authors have adequately addressed your comments raised in a previous round of review and you feel that this manuscript is now acceptable for publication, you may indicate that here to bypass the “Comments to the Author” section, enter your conflict of interest statement in the “Confidential to Editor” section, and submit your "Accept" recommendation.

Reviewer #1: All comments have been addressed

Reviewer #2: All comments have been addressed

2. Is the manuscript technically sound, and do the data support the conclusions?

Reviewer #1: Yes

Reviewer #2: (No Response)

3. Has the statistical analysis been performed appropriately and rigorously? 

Reviewer #1: Yes

Reviewer #2: (No Response)

4. Have the authors made all data underlying the findings in their manuscript fully available?

Reviewer #1: Yes

Reviewer #2: (No Response)

5. Is the manuscript presented in an intelligible fashion and written in standard English?

Reviewer #1: Yes

Reviewer #2: (No Response)

6. Review Comments to the Author

Reviewer #1: I want to thank the authors for their in depth rebuttal letter and their revision adressing all my concerns! Great work!

Reviewer #2: (No Response)

7. PLOS authors have the option to publish the peer review history of their article (what does this mean?). If published, this will include your full peer review and any attached files.

Reviewer #1: **Yes: **Tobias Appel

Reviewer #2: **Yes: **Mazyar Fallah

---

## [Editor Report · Acceptance letter]

25 Jul 2022

PONE-D-22-08472R1 

Seeing the Forrest through the trees: Oculomotor metrics are linked to heart rate 

Dear Dr. Hoogerbrugge:

I'm pleased to inform you that your manuscript has been deemed suitable for publication in PLOS ONE. Congratulations! Your manuscript is now with our production department. 

Kind regards, 

on behalf of

Prof. Dr. Enkelejda Kasneci 

Academic Editor

PLOS ONE